# Blood-Based Markers for Skeletal and Cardiac Muscle Function in Eventing Horses before and after Cross-Country Rides and How They Are Influenced by Plasma Volume Shift

**DOI:** 10.3390/ani13193110

**Published:** 2023-10-05

**Authors:** Johanna Giers, Alexander Bartel, Katharina Kirsch, Simon Franz Müller, Stephanie Horstmann, Heidrun Gehlen

**Affiliations:** 1Equine Clinic, Internal Medicine, Freie Universität Berlin, Oertzenweg 19b, 14193 Berlin, Germany; 2Institute for Veterinary Epidemiology and Biostatistics, Freie Universität Berlin, Königsweg 67, 14163 Berlin, Germany; alexander.bartel@fu-berlin.de; 3Department Sensors and Modeling, Leibniz Institute for Agricultural Engineering and Bioeconomy (ATB), Max-Eyth Allee 100, 14469 Potsdam, Germany; kkirsch@atb-potsdam.de; 4Laboklin Veterinary Laboratory Diagnostics, Steubenstrasse 4, 97688 Bad Kissingen, Germany; sf.mueller@laboklin.com; 5German Olympic Committee for Equestrian Sports (DOKR), Freiherr-von-Langen-Straße 15, 48231 Warendorf, Germany; shorstmann@fn-dokr.de

**Keywords:** eventing, cross-country, exercise, recovery, sport horses, cardiac muscle, cardiac troponin I (cTnI), performance diagnostics, equine sport physiology

## Abstract

**Simple Summary:**

Horses participating in cross-country tests are subjected to high physical stress. In this study, the blood values of 20 horses were examined before and after cross-country rides. The aim was to determine whether blood-based markers of muscle function change after cross-country rides. Parameters that provide information about fluid balance, muscle enzymes, metabolites and cardiac muscle-specific markers were investigated. We developed an approach to eliminate the concentration changes caused by shifts in fluid balance. Parameters were measured before, 10 and 30 min after exercise and the next morning and were evaluated using a mixed model. At 30 min post exercise, most parameter concentrations changed in an exercise-dependent manner. The next morning, most exercise-dependent markers recovered rapidly; creatine kinase (CK) (26% increase; *p* = 0.008) and lactate dehydrogenase (LDH) (15% increase; *p* < 0.001) showed a declining but sustained increase. Cardiac troponin I (cTnI) increased above the reference range in 40 of the 55 rides (73%) and in 18 of 20 horses in the morning after exercise.

**Abstract:**

Horses competing in cross-country tests are subjected to high physical demands. Within the scope of this prospective longitudinal study, blood values of 20 elite eventing horses were examined before and after two- to four-star cross-country rides. The aim was to find out whether blood-based markers for skeletal muscle and cardiac muscle function change after cross-country exercise. Parameters that provide information about fluid balance, muscle enzymes, metabolites and cardiac muscle-specific markers were investigated. We developed an approach to eliminate the concentration changes caused by reduced plasma volume. Parameters were measured pre, 10 and 30 min post exercise and the next morning and were evaluated using a mixed model. Thirty minutes after exercise, most parameter concentrations changed in an exercise-dependent manner. The next morning, most exercise-related markers recovered rapidly, while creatine kinase (CK) (26% increase; *p* = 0.008) and lactate dehydrogenase (LDH) (15% increase; *p* < 0.001) showed a declining but sustained increase. Cardiac troponin I (cTnI) increased above the reference range in 40 of the 55 rides (73%) and in 18 of 20 horses in the morning after exercise.

## 1. Introduction

In recent years, the equestrian discipline of eventing has received increased attention due to growing concerns about the welfare of horses in cross-country competitions. These demanding competitions require horses to jump over solid obstacles at high speed, putting them under considerable physical strain. To reduce the risk of injury and ensure the health and performance of the horses, it is essential to assess musculoskeletal and cardiac function during the recovery period.

In human high-performance sports, compiling a panel of exercise-dependent biomarkers and establishing individual reference ranges has helped to achieve the assessment of recovery status [1,2,3,4,5,6,7,8,9].

In horses, many different blood-based markers have recently been investigated for their suitability as exercise-dependent markers of skeletal and cardiac muscle function. Plisak et al. [10] investigated the previously less-studied cytokines IL-1ra and IL-13 in comparison with the well-known anti-inflammatory cytokine interleukin 10 (IL-10) in serum concentrations during endurance and racing in horses of different fitness levels. Kirsch and Sandersen [11] examined various electrolytes in eventing horses before and after cross-country rides to analyze the acid–base balance.

In human elite sports, the specificity of a parameter for a particular metabolic process is considered less relevant than the strong dependence on exercise to qualify the parameter for performance diagnostics [1,2,3,4,5,6,7,8]. Therefore, this study examined parameters that are part of the standard repertoire of any laboratory and whose functions have already been well studied.

The enzymes creatine kinase (CK), lactate dehydrogenase (LDH) and aspartate aminotransferase (AST) have long been proven to diagnose exercise-induced myopathies [12] and have also been used in many studies to assess skeletal muscle function in eventing horses before and after cross-country rides [13,14,15,16,17]. In humans, it is recommended to measure the enzymes CK, LDH and AST to obtain information on muscle cell damage [7,12,18].

One parameter that is considered a specific indicator of myocardial damage is cardiac troponin I (cTnI). However, post-exercise increases in cTnI levels are common in active horses and must be distinguished from pathological increases, as pointed out by Lippi and Sanchis-Gomar [19]. In fact, cTnI only enters circulation when myocardial cells are under increased stress [18,20,21,22,23,24,25]. Studies on endurance horses [26,27,28] reported an increase in cTnI levels after endurance rides, while Fazio et al. [29] found cTnI increases after jump training in healthy horses. Ayvazoglu et al. [30] and Pourmohammad et al. [31] found cTnI increases in Arabian and thoroughbred racehorses after training. In the context of racehorses, Rossi et al. [32] demonstrated that increased cTnI concentrations and cardiac arrhythmias after racing indicate myocardial damage. Gunther-Harrington et al. [33] also described exercise-induced cTnI elevation in thoroughbred racehorses receiving furosemide. To our knowledge, this is the first study to investigate whether cTnI also increases in eventing horses after cross-country exercise.

In this study, we examine whether blood-based markers for the skeletal and cardiac muscle function of top-class eventing horses change post exercise under realistic and updated competition conditions. Unlike previous studies that overlook the shift in plasma volume when analyzing blood parameters post exercise, we propose an adjustment that enhances the interpretation of blood parameters immediately after exercise.

## 2. Materials and Methods

Riders and owners received written information about the study conditions and agreed in writing to voluntary and unremunerated participation. The study was registered with the regulatory state office of Berlin (1-02.04.40.2022.VG006) but was not classified as an animal experiment. Blood samples were taken by veterinarians who were involved in routine performance diagnostic care through the ‘performance monitoring program’ of the German Olympic Committee for Equestrian Sports (DOKR) project. All horses were examined clinically and via echocardiography by veterinarians of the DOKR project and declared healthy before participation.

### 2.1. Study Design 

A total of eight riders with 20 horses participated in the study, which took place at 14 international 2- to 4-star-level eventing competitions at 5 different venues in Germany and Poland between March and September 2022.

All horses were sampled at two to five competitions during the season, resulting in a sample count per horse ranging from 4 to 20. For each competition, the horses went through the mandatory veterinary checks before and after the cross-country test, which declared the horses ‘fit to compete’.

#### 2.1.1. Horses

Subjects for the present longitudinal observational study were horse–rider combinations that are monitored by the ‘performance monitoring program’ of the German Olympic Committee for Equestrian Sports (DOKR) project.

The horses participating in the study were between 7 and 15 years old, with a mean age of eleven years. The population consisted of ten mares and ten geldings. The horses belonged to nine different warmblood breeds (Table 1). Relevant information regarding the identification of the horses was obtained from FEI database [34].

#### 2.1.2. Riders

The riders were aged between 21 and 39, with an average age of 28. The study included three male and five female riders. To compete at the respective test level, all riders achieved the corresponding performance levels through placings in lower classes, ensuring that they had comparable performance levels.

#### 2.1.3. Training Schedules

The horses were individually prepared for competition by their riders and trainers for the corresponding competition requirements. Each training plan includes dressage, jumping and cross-country training, as well as basic endurance training, days with light work and days with only paddock or pasture. The specific training schedule of each horse was neither reported nor standardized.

#### 2.1.4. Exercise

An eventing competition comprises the three sub-tests dressage, jumping and cross-country. This study focuses on the cross-country test, which, depending on the test format, was either the second or the third test in the eventing competition.

There were 9 rides sampled at a two-star level, 31 rides at three-star level and 15 rides at four-star level, following the rules of the Fédération Equestre Internationale (FEI) for international eventing competitions [35]. Specific test requirements for the cross-country tests can be found in Table 2. The average score achieved in the cross-country test was 9.2 penalty points. In two rides, riders were eliminated due to horse falls, and in one ride, the rider retired. Information regarding nutrition, water intake and additional exercise during the competition days was not recorded.

#### 2.1.5. Sampling Times

Sampling times included the morning before exercise (Pre), ten minutes (10 min) and thirty minutes (30 min) after the end of exercise and 24 h after Pre-sample (next morning). Samples pre-exercise and the next morning were taken between 04.00 a.m. and 7.30 a.m. Due to varying start times, the 10 min and 30 min samples were collected between 09:00 a.m. and 05:00 p.m. Depending on the horses’ start times, the time intervals between the Pre-samples and the samples shortly after exercise, as well as between the 30 min samples and the next-morning samples, varied. The time span between 30 min post exercise and the next morning ranged from 11 to 21 h, while the time span between Pre and next morning was 24 h (±1 h).

### 2.2. Sample Collection 

Blood samples were obtained from the horses’ jugular veins, with the puncture site disinfected with 1-propanol. Venous blood sampling was performed with a Vacutainer system using 20 G needles and PET (polyethylene terephthalate) tubes. Following each blood collection, EDTA whole blood tubes were immediately refrigerated at +5 °C, while serum gel tubes were maintained at room temperature for 30 to 60 min until clotting was complete. A portable centrifuge, type EBA 200 from Andreas Hettich GmbH & Co. KG (Tuttlingen, Germany), was used to centrifuge serum tubes at 1000× *g* for 10 min before the serum was transferred to uncoated plastic tubes and immediately frozen at −20 °C.

#### 2.2.1. Sample Storage and Transport to the Testing Laboratory 

All samples were refrigerated (+5 °C) or frozen (−20 °C) within minutes of processing at the event site. Lactate values were determined using a BIOSEN system, and sample cups were stored at +5 °C until analysis. All samples were then transported within 48 h from the event site to the in-house laboratory of the German Equestrian Federation (FN) in Warendorf, Germany, where refrigerated EDTA whole blood and BIOSEN lactate samples were analyzed immediately. The serum samples were stored at −20 °C for a maximum of 8 weeks before being transported frozen to the external testing laboratory, LABOKLIN GmbH & Co. KG (Bad Kissingen, Germany).

#### 2.2.2. Blood Parameters

Blood parameters specific to cardiac and skeletal muscle, including enzymes and metabolites, were measured, along with parameters providing information about shifts in fluid balance (Table 3). Red blood cells (RBC), hematocrit (HCT), total protein (TP), albumin (Alb), urea and symmetrical dimethylarginine (SDMA) were determined at all measurement time points. Aspartate aminotransferase (AST), creatine kinase (CK), lactate dehydrogenase (LDH), creatinine (CREA), calcium, chloride, potassium, magnesium, sodium, inorganic phosphate (Inorg. Phosphate) and cardiac troponin I (cTnI) were not determined 10 min post exercise, but were at all other measurement time points.

#### 2.2.3. Measurement Techniques

Hematology was assessed using laser flow cytometry and laminar flow impedance in a ProCyte DX hematology analyzer from IDEXX Laboratories Inc. (Westbrook, ME, USA). Blinded serum samples were further processed in the investigation laboratory (LABOKLIN, Bad Kissingen, Germany). cTnI was measured through a chemiluminescence assay (LIA) using an ADVIA Centaur XPT 2000 (Siemens, Munich, Germany). All other blood chemistry parameters were measured photometrically (PHO) or potentiometrically (POT) with a Cobas 8000 analyzer (Roche, Basel, Switzerland).

### 2.3. Missing Values

Sample sets were complete in 54 of 55 rides. In one set, the ‘next morning’ sample is missing because the rider refused to take the sample in a stressful situation. In two blood collections, the serum tube was lost after collection. In one blood collection, the amount of blood was insufficient to fill all sample tubes, resulting in the absence of certain values for this measurement. Additionally, 34 lactate values are missing because no lactate sample cups were filled at the ‘pre’ and ‘next morning’ time points at the beginning of the season.

### 2.4. Data Analysis

All statistical analyses were performed using Jamovi (version 2.3.21.0) [36]. Normality of data was visually assessed using box plots, histograms and Q–Q plots. Due to non-normality, the parameters lactate, AST, calcium, CK, magnesium and potassium were log-transformed. Cardiac Troponin I (cTnI) values were log1p-transformed.

#### 2.4.1. Plasma Volume Loss Adjustment

Blood values are reported as concentrations, inherently defined as amount per volume. During exercise, not only the quantity of the blood parameter (cell, enzyme, metabolite) changes but also the volume due to the three effects of splenic contraction, plasma volume loss and dehydration. This means that changes in the quantity of the blood parameter can be masked or simulated. To understand the changes in individual blood parameters after exercise and draw conclusions about the function of specific systems such as the skeletal muscles, it can be helpful to separate volume effects from quantity effects.

To account for dehydration following endurance training in Arabian horses, Witkowska-Pilaszewicz et al. [37] adjusted SAA values using total protein. However, TP may not be an accurate measure for assessing the degree of intravascular plasma volume loss in horses after maximal exercise, as splenic contraction can also release additional proteins (particularly globulins) into the bloodstream, as seen in Masri et al. [38].

Therefore, we propose using albumin as a marker of intravascular plasma volume loss. An increase in albumin concentration during or shortly after submaximal exercise is primarily due to plasma volume shifts [39,40]. We adjusted values at 10 min and 30 min post exercise, while Pre-values remained unadjusted. Next-morning values were also not adjusted, assuming that albumin levels in the morning after exercise are not related to plasma shift during exercise. Each individual 10 min and 30 min value was adjusted using the formula:Adjusted value = Measured value _time point x_/(1 + plasma volume loss (%) _time point x_)(1)

#### 2.4.2. Mixed Model

Estimated marginal means (EMM) for each time point and parameter were calculated using a mixed model. Time Point was used as fixed effect, while “horse” nested in “rider”, and “level” nested in “competition” were used as cluster variables. The normality of the residual distribution was checked using a normal likelihood plot of the residuals.

The mixed model was applied to both the observed and adjusted data sets. Estimated marginal means (+confidence intervals) of adjusted values are represented in figures as light gray dots with dark gray whiskers. In this study, 95% confidence intervals were calculated, and the significance threshold was set at 5%.

*p*-values given in the figures refer to change in EMM between ‘Pre’ and the respective time point and were calculated using the plasma shift-adjusted values. In the case of albumin, *p*-values were calculated using the non-adjusted values. The estimated marginal means for the adjusted values of albumin show a straight line between Pre and 30 min after exercise, which illustrates the effects of the adjustment. For cTnI, no adjusted EMM are shown as they do not differ from EMM without adjustment. The *p*-values for changes between time points in cTnI are also identical with and without adjustment.

We examined whether the tested parameters exhibited exercise-dependent increases or decreases, the magnitude of these changes and how far their estimated marginal means deviated from reference ranges for healthy horses at rest. Relative changes in parameter EMM_unadjusted_ across the tested time points were calculated. Blood values exceeding the reference ranges are commonly highlighted in lab reports, enabling quick identification by veterinarians. For our study we’ve evaluated the frequency of these outlying results post-ride to determine if such deviations are common or potentially pathological.

## 3. Results

The observed descriptive data (Appendix A) as well as the estimated marginal means of the unadjusted values and their 95% confidence intervals (Appendix A) for the respective parameters are presented in the Appendix A.

### 3.1. Fluid-Balance-Related Parameters

Based on the increase in albumin concentration up to 30 min after exercise, we calculated a reduction in intravascular plasma volume of 9.5% 10 min and 7.1% 30 min after the end of exercise using the reported mixed model (Figure 1a).

All fluid-balance-related parameters peaked at the earliest measured time point post exercise (unadjusted) (Figure 1a–h). Exceptions were urea, which slightly increased until the next morning, and SDMA, with peaked 30 min after the end of exercise (unadjusted and adjusted) (Figure 1e,f). Thirty minutes after the end of exercise, observed RBCs were 25% and HCT was 28% higher than before exercise (unadjusted). In the morning after the cross-country test, there were deviations of no more than 6% in all fluid-balance-related parameters compared with “Pre” exercise.

### 3.2. Muscle Enzymes

The maximum value for CK was 537 U/L, measured 30 min after exercise. CK reached values above the reference in 47% of the rides, and the CK changes were significant due to exercise (*p* < 0.05). Thirty minutes after the end of exercise, CK levels were above the reference range (unadjusted) (Figure 2a). CK levels 30 min after exercise were 45% higher and the next morning they were 26% higher than those pre-exercise (unadjusted).

LDH was 676.6 U/L in maximum, measured the next morning after cross-country. In 73% of the rides, LDH increased above the reference range. LDH changed significantly due to exercise (*p* < 0.05). Thirty minutes after the end of exercise and the next morning, LDH levels were above the reference range (unadjusted) (Figure 2b). LDH levels were 20% higher 30 min after exercise and 15% higher the next morning than before exercise (unadjusted).

Maximum AST was 512.2 U/L, measured the next morning after cross-country. AST reached values above the reference range in 44% of the rides. AST exceeded the current reference range only 30 min after exercise, if plasma volume loss is not considered. If plasma volume loss is considered, AST does not change between the time points of the measurement (*p* = 0.934) (Figure 2c).

### 3.3. Muscle Metabolites

The maximum measured lactate value was 27.92 mmol/L, measured 10 min post exercise. Lactate was above the reference range in 100% of the rides, at least at the earliest measured time point post exercise. Lactate changed significantly due to exercise (*p* < 0.05). The lactate levels 10 min after the end of exercise were 14 times (1293% increase) and 30 min after exercise still 5.5 times (453% increase) higher than before exercise (unadjusted) (Figure 3a).

The range of measured Creatinine (64 to 167 µmol/L) values was close to the reference range. Creatinine changed due to exercise (*p* < 0.05). Creatinine was 31% higher 30 min post exercise and 5% higher the next morning than before exercise (*p* < 0.01) (unadjusted) (Figure 3b).

Calcium values ranged between 2.7 and 3.3 mmol/L, and the changes due to exercise were significant (*p* < 0.05). Unadjusted calcium showed an increase of 3% 30 min after the end of exercise (unadjusted) (Figure 3c), but the adjusted values were lower 30 min after exercise than pre-exercise.

Magnesium values ranged from 0.5 to 0.9 mmol/L. Magnesium changed due to exercise (*p* < 0.05). Unadjusted magnesium was 3% lower 30 min after the end of exercise (unadjusted) (Figure 3d).

Inorganic phosphate was measured between 0.4 and 1.8 mmol/L. The changes were significantly due to exercise (*p* < 0.05). Inorganic phosphate displayed values below the reference range in 11 rides (20%; *n* = 55) 30 min post exercise. Inorganic phosphate showed a decrease of 26% 30 min post exercise (unadjusted) (Figure 3e).

The range of measured potassium was 1.8 to 7.9 mmol/L. Potassium was below the reference range in 16 rides (29%; *n* = 55) at different time points. Potassium showed no significant change due to exercise (*p* = 0.485).

The next morning, only deviations of no more than 2% from the pre-value were measurable in the electrolytes calcium, magnesium, Inorg. phosphate and potassium.

### 3.4. Cardiac Markers 

For all horses examined, cTnI values above the reference range were measured in 40 of 55 rides (73.0%). Mean cTnI concentrations increased continuously between “Pre” and “next morning”. With an estimated marginal mean of 0.07 ng/mL (0.04; 0.09 ng/mL), next-morning values exceeded the reference range (<0.03 ng/mL) (Figure 4).

The highest troponin value measured was 0.41 ng/mL and was measured the morning after a cross-country test. The highest value measured prior to start was 0.19 ng/mL and 0.25 ng/mL 30 min after end of the exercise. The maximum values belonged to one horse in one ride at the beginning of the season. This horse had troponin I values above the reference range in eight of eight samples. The average percentage increase between pre-value and next-morning value is the same for this horse as for the other horses (2.5-fold increase), with a mean delta cTnI between “Pre” and “next morning” of 0.15 ng/mL. Clinically and echocardiographically, no signs of heart disease were detected in this horse before and during the season, prior and after each competition. In the cross-country tests, this horse’s score ranged from 0.0 to 9.2 penalty points.

## 4. Discussion

Muscle enzymes, muscle metabolites and cardiac markers show exercise-dependent changes, with and without considering shifts in plasma volume. Shortly after exercise, both muscle enzymes and muscle metabolites show deviations from the pre-values. In the morning after exercise, the muscle enzymes CK and LDH are still significantly elevated, but compared with 30 min post exercise they decreased already. There is a significant cTnI increase in eventing horses following cross-country exercise.

### 4.1. Fluid Balance

There are three major effects that influence fluid balance during and after exercise.

First, splenic contraction releases additional red blood cells into the bloodstream to increase oxygen-carrying capacity, which increased HCT during exercise up to 10% in intact and splenectomised horses in an incremental treadmill exercise test [41]. Even though the cross-country ride represents a submaximal exercise, a 17.9% HCT increase ten minutes after exercise was observed in the eventing horses of the present study, suggesting that either the capacity of the storage spleen is even greater than estimated by McKeever et al. [41], or that the HCT was additionally increased by plasma volume loss. Since plasma volume loss was estimated to be 9.5% (10 min post exercise) using albumin, which is little affected by splenic contraction [41], the difference in HCT increase is 8.4%, consistent with the literature data on splenic contraction [38,42].

Second, there is a plasma volume shift from intravascular to extravascular space because hydrostatic pressure increases during exercise [38,40,42,43]. In the present study, TP and Alb increased during exercise by about as much as Masri et al. [38] and McKeever et al. [42] calculated for intravasal plasma volume loss during maximal exercise 30 years ago. Masri et al. [38] calculated a 13% plasma volume loss within the first ten minutes after the end of 1000 m maximal exercise using blood chemical values like plasma protein and electrolytes. McKeever et al. [41,42] calculated a plasma volume loss of 5 to 10% during exercise in splenectomised and intact horses. While most descriptive studies do not adjust for plasma volume shifts during exercise [40,43,44], Witkowska-Pilaszewicz et al. [37] adjusted SAA values by using total protein. The adjustment of individual values in an observational study may give the impression of a distorted representation of the data. In our opinion, the applied plasma volume shift adjustment is, rather, a plasma volume standardization that allows for an unmasked view of the changes in parameter amounts. It seems reasonable to take massive fluid shifts into account, especially if you want to understand equine physiology after exercise.

Third, fluid loss (dehydration) occurs through sweating and breathing [40,45]. Electrolyte values (sodium, chloride), adjusted for plasma volume shift shortly after exercise, reflect this electrolyte loss [44,45]. Compared with endurance competitions, sodium and chloride decreases are less relevant in eventing horses, which could be due to the substantially shorter exercise duration of eventing rides. But the adjustment for plasma volume shift reveals that the concentrations do change and that the horses must compensate for these shifts.

To account for these three effects, the adjustment was developed, and the *p*-values were recalculated. Since the purpose of the adjustment was to compensate for albumin changes, and albumin is the main component of the TP, it was reasonable to conclude that the adjusted EMM of albumin and TP 10 min and 30 min after the end of exercise were unchanged compared with Pre, and the changes were not statistically significant (*p* > 0.05).

The slight increase in renal values of UREA and SDMA could be explained by the increased production of waste products during exercise from the skeletal muscles and due to the increased metabolism [44]. In addition, renal excretion could be slightly reduced due to plasma shift and fluid loss during exercise compared with the resting state of the horse.

While SDMA concentration in the study by Riccioni et al. [46] was significantly reduced by physical exercise, which speaks for a reduced NO-forming capacity, Nyborg et al. [47] showed increased SDMA concentrations after extreme marathons in humans, which were considered a risk factor for arteriosclerosis. In the present study, neither positive effects of exercise on mean values nor alarming individual values were observed.

The analysis of the fluid balance parameters shows the influence of the physiological effects of spleen contraction, plasma volume shift and dehydration. Since the concentrations of plasma solutes can be influenced by plasma volume loss [40,42,44], an adjustment was made.

There were exercise-dependent changes in the fluid balance of eventing horses following cross-country exercise. Compared with other studies using maximal high-intensity exercise [38,40,41,42] our study found similar effects due to splenic contraction. Compared with studies on endurance rides [37,43], we also found evidence of plasma volume shifts during cross-country exercise, but the amount of fluid loss through sweating and respiration is likely to be less in eventing horses because of the much shorter duration of exercise.

### 4.2. Muscle Enzymes

AST did not increase significantly due to exercise. However, trained eventing horses have higher AST values than clinically healthy horses, which were used to create the reference ranges. On EclinPath [39], it is described that highly trained horses have up to 30% higher AST serum activity at rest and that resting levels in early training are even 50 to 100% higher than those in horses without training [39]. Therefore, resting AST levels could be an interesting factor in detecting training effects in a longitudinal evaluation. Reasons for the increased AST baseline values might be continuous muscle stress due to physical exertion. This is contradicted by the fact that the physical stress varies, but the AST values were elevated in a very stable manner. Another possible reason for the increased AST baseline values might be an overall increased muscle metabolism, which could be caused by larger muscle mass or an altered composition of the muscle mass, respectively, as well as an altered metabolism of the muscle mass.

CK showed a significant maximum increase to 215 U/L (mean) 30 min after the end of exercise. Almost 30 years ago, when eventing competitions were structured differently and training monitoring was not yet professionalized, CK values of 1465.8 U/L on average were measured 10 min after the end of the cross-country phase [17]. At that time, the ‘roads and tracks’ and the ‘steeplechase’ phase still existed, which made the total distance of cross-country rides substantially longer [16]. This shows that the muscle fiber damage of eventing horses seems to have decreased significantly over the decades, which could be due to the reduction in test requirements on the one hand and the significantly improved training monitoring and exercise scheduling on the other hand.

Thoroughbred racehorses, examined by Arfuso et al. [48], had comparable CK values and lower LDH values 30 min after 1200 m to 2000 m gallop races, which may indicate a higher training status in these horses. However, in contrast to the cross-country course, the length of the racetrack is shorter, the speed is higher and there are no jumps.

LDH values 30 min after exercise in the present study (LDH_30 min_: 437 (395;479) U/L) were about the same as those of trained purebred Arabian racehorses (417 ± 28.20 U/L) in daily training (2000 m gallop) evaluated by Kowalik and Tomaszewska [49]. The next morning, LDH levels in the present study did not drop to pre-values, which was the case in the purebred Arabian racehorses [49].

Lactate values were lower in the purebred Arabian racehorses (immediately after exercise: 4.64 ± 3.14 mmol/L) than in the cross-country horses (10 min post exercise 8.53 U/L (4.89–12.00 mmol/L)).

Since LDH is responsible for the lactate breakdown in the cell [50], it is reasonable to assume that the prolonged LDH accumulation is related to the higher amount of lactate. However, it should be considered that the higher amount of lactate increases the LDH activity, but not necessarily the measurable amount of LDH in the blood. This could also be due to a prolonged increase in cell membrane permeability. The comparability of the two studies is nevertheless questionable, as exercise differed between the studied rides.

CK and LDH showed significant increases the next morning, although the enzyme levels decreased compared with the 30 min values.

The maximum CK and LDH values measured in the present study (CK 537 U/L; LDH 676.6 U/L) are quite far from the threshold values of exercise-induced myopathies. In cases of tying up syndrome, which is the mildest described form of exercise-induced myopathies, CK values up to 2000 U/L and LDH values up to 1500 U/L are measured [12].

To evaluate the sustained increase in the muscle enzymes CK and LDH in the morning after exercise, the slow clearance should be taken into account, which corresponds to a half-life of 9.4 ± 5.7 h for CK [51] and a half-life of 7.65 h [52] for LDH. Accordingly, the slight persistence of CK and LDH in the blood is attributable to slow excretion rather than sustained release.

The muscle enzymes CK and LDH show exercise-dependent increases above the limit of the reference range following cross-country exercise. However, compared with exercise-induced myopathies, other equine disciplines and former eventing competitions, the percentage increases are small, and values already decrease within one night. The slight persistence of CK and LDH in the blood is attributable to slow excretion rather than sustained release.

### 4.3. Muscle Metabolites

Muscle metabolites respond differently; lactate shows strong increases but does not appear to be associated with severe muscle fiber damage, as muscle enzymes show only small increases.

Creatinine may originate from the muscles [53,54,55,56], as creatinine is a side product of an energy-providing reaction in the muscles. McKeever et al. [57] and Hinchcliff et al. [58] found that the glomerular filtration rate and filtration fraction do not change in horses during submaximal exercise. Thus, the creatinine increase observed in our study is most likely not due to altered renal function.

All muscle metabolites except potassium were significantly altered by exercise. Electrolytes such as calcium, magnesium, inorganic phosphate and potassium may be lost through sweat [59,60,61]. The changes in the amount of electrolytes can also reflect adaptions in the acid–base balance. The concentrations of electrolytes like potassium are physiologically strongly regulated and deviations in these parameters in individual animals should attract attention, as they may be signs of severe exertion.

Inorganic phosphates drop 30 min after exercise. An accumulation of inorganic phosphates in the muscles, which seems to contribute to peripheral fatigue [62], could be related to a drop in concentration in the blood, although only small amounts of inorganic phosphate circulate in the blood [62,63,64,65]. It could be hypothesized that the plasma dissolved inorganic phosphate is used to produce Adenosine-Triphosphate (ATP) via phosphorylation and the breakdown products then accumulate in the muscle cells, causing fatigue [63]. However, increased muscular work under stress is known to cause acidification, which would spill over into the bloodstream if not buffered by physiological systems [66]. The inorganic phosphate plays a role here, as it binds H^+^ in the proximal renal tubules, thus changing from HPO_4_^−^ into H_2_PO_4_, which cannot be absorbed and is excreted, contributing to the buffering of the accumulating H^+^ ions [66].

The significant increase in lactate levels with a minimal increase in muscle enzymes suggests that the horses tested showed no signs of muscle damage even though large amounts of lactate accumulated. Blood lactate levels reflect a delicate balance between lactate production in muscles and lactate consumption by oxidative metabolism, particularly in oxidative muscle fibers [66]. The current view is that lactate is not simply a waste product of anaerobic metabolism [50]. In trained individuals, both humans and horses, lactate serves various functions in aerobic metabolism during periods of exercise [50]. However, the accumulation of lactate in the blood indicates that the proportion of anaerobic energy production in the muscles is significantly increased and exceeds the capacity for lactate consumption [66]. Nevertheless, horses appear to be able to withstand this stress, possibly due to training, without evidence of muscle damage.

Muscle metabolites show different responses post exercise. The increases in lactate and creatinine are attributable to the increased muscle metabolism. The changes in electrolyte amounts only become apparent through plasma shift adjustment. Recovery was such that all muscle metabolites were restored to baseline levels by the following morning in the majority of horses studied.

### 4.4. Cardiac Markers

The increase in cTnI indicates cardiac fatigue in eventing horses after cross-country riding due to the increased release of cTnI from cardiomyocytes [20,21]. The reasons for this, besides myocardial damage, could be increased cell membrane permeability because of changes in oxygen tension or local pH during exercise [67,68].

To identify myocardial damage, measurements at several time points after exercise are necessary to assess how long the cTnI release lasts. Sustained cTnI release over several days is indicative of myocardial damage, while rapidly decreasing cTnI release is indicative of a physiological process [21,24]. In humans, the literature reports a release pattern in which the rise and peak occur within the first four hours after physical exercise [21,24]. If myocardial necrosis occurs, the peak occurs later (about 8 h post exercise), and the decline does not occur in the first 24 h but over several days [21,24]. A distinction between a physiological increase and myocardial necrosis is not possible in this study setting, as the peak cTnI value is expected at least four hours after exercise, and in the present study, no measurements could be taken between 30 min after exercise and the next morning.

The maximum value of 0.41 ng/mL measured in this study is higher than the maximum value of 0.00964 ng/mL measured in clinically healthy racehorses 2 h after a race by Rossi et al. [32].

In endurance horses after 120–160 km races, Ertelt et al. [28] measured a maximum value of 8.2 ng/mL, which is more than 20-fold higher than the maximum value measured in this study.

Flethøj et al. [43] calculated a mean cTnI value of 0.010 ng/mL 45 min to 3 h post race and 0.005 ng/mL the following morning (10 to 14 h post race) in endurance horses.

However, in Rossi et al. [32] no sample was taken later than 2 h after the race and in Flethøj et al. [27], Rossi et al. [32] and in our study, the peak was not measured after 4 h.

Only Gunther-Harrington et al. [33] measured 4 h after the race in racehorses and reported a range of 0.00 to 0.07 ng/mL and 0.00 to 0.05 ng/mL for 24 h after the race, which are therefore comparable to the values measured the next morning in this study.

The different maximum values and ranges reported by Rossi et al. [32], Ertelt et al. [28], Flethøj et al. [43] and our study may be related to the time of measurement and the type of exercise.

The maximum cTnI values of 0.19 ng/mL before, 0.025 ng/mL 30 min after and 0.41 ng/mL the next morning observed in one horse at one competition at the beginning of the season in our study exceeded the reference ranges for healthy horses at rest and after exercise, summarized by [69]. The ranges for cTnI compiled by [69] for horses with heart disease are very wide and include the maximum values measured in the present study. However, values are given for horses with heart disease that exceed the maximum measured in this study by a factor of 10 to 100.

Trachsel et al. [70] measured lower maximum cTnI values in healthy horses and horses with mitral valve insufficiencies in a submaximal exercise test on a treadmill than those in the current study. This supports the assumption that more release of cTnI, i.e., greater cardiac fatigue, occurs when anaerobic exercise is performed. Also, Eijsvogels et al. [71] provide evidence that the magnitude of the increase in cTnI is to some extent related to exercise intensity.

In summary, an exercise-dependent increase in cTnI concentration was observed in eventing horses. These elevations may be physiological [69,72], but should induce appropriate follow-up examinations, especially if accompanied by symptoms such as weakness, poor performance, dyspnea or syncope [69,73].

### 4.5. Muscle and Cardiac Fatigue

Fatigue is formally defined as a state in which the capacity of striated muscles to generate force diminishes under sustained stimulation [74]. However, it is important to note that fatigue extends beyond striated skeletal muscles and is a multifaceted phenomenon influenced by both peripheral and central factors [75,76,77]. Cardiac fatigue, on the other hand, is characterized by transient alterations in heart muscle performance that may be linked to myocardial cell damage [78]. In our current context, cardiac fatigue is classified as pathological only when accompanied by evidence of myocardial necrosis [24].

The increase in cTnI levels observed following cross-country exercise reflects changes in the cellular function of cardiomyocytes, which we posit aligns with the concept of cardiac fatigue. Notably, the definition of cardiac fatigue does not inherently necessitate the presence of myocardial necrosis, a finding that was not detected in our study. Consequently, we interpret the elevated cTnI levels we measured as an indication of a physiological manifestation of cardiac fatigue.

### 4.6. Limitations

To really assess the kinetics of different blood parameters, especially cTnI, several blood samples at later time points would have been necessary [79]; however, this was not feasible for organizational reasons and not justifiable for animal welfare reasons.

The varying time span between 30 min post exercise and the next morning (11 to 21 h) is likely to influence the results. The kinetics of muscle enzymes and cTnI suggest higher levels 11 h after exercise than 21 h after exercise.

The selection of blood parameters was limited to established parameters. This means that the organ systems of the muscle and heart were not examined in an all-encompassing manner [18,69]. A limitation of this study was the number of horses, which was too small to establish reference values according to the American Society for Veterinary Clinical Pathology guidelines for determining reference intervals in animal species [80]. A major limitation of the study is that the statistical evaluation does not go beyond the level of the mean values and, thus, is not capable of identifying the horses that deviate from these mean values. Nevertheless, this study provides an important basis for an individualized assessment.

## 5. Conclusions

In conclusion, relevant changes in almost all muscle enzymes and muscle metabolites occurred within 30 min after the cross-country test. After one night of recovery, only urea, CK, LDH and cTnI were significantly increased, while the other parameters returned to pre-exercise values. The adjustment based on albumin concentration conclusively elucidated the impact of plasma volume loss on changes in blood-based markers, as demonstrated in our study. The reported cTnI increase suggests the presence of cardiac fatigue in eventing horses, but myocardial damage was not detected. To assess whether cTnI increases indicate a pathological state, multiple post-exercise cTnI measurements and echocardiographic examinations, including myocardial function measurements, are required.

## Figures and Tables

**Figure 1 animals-13-03110-f001:**
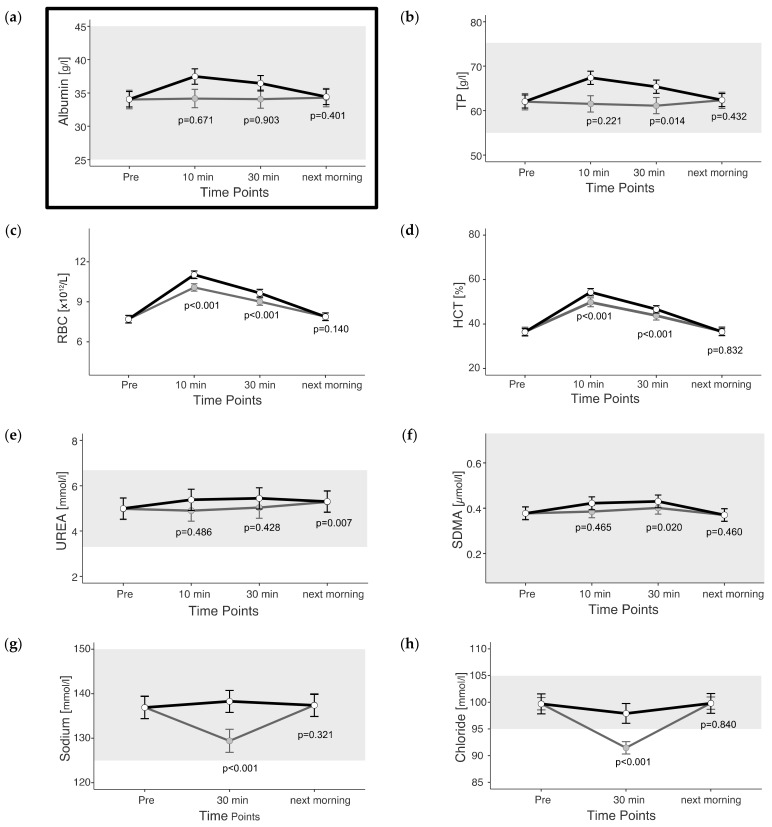
(**a**–**h**) Estimated marginal means (EMM) (white dots) and their 95% confidence intervals (black whiskers) of fluid balance blood parameters at measured time points. Gray dots and whiskers represent EMM and 95% confidence intervals when 10 and 30 min post-exercise values were adjusted for plasma shift. *p*-values apply to the difference between Pre and the respective time point. Y-axis is scaled according to the observed scores. The gray box symbolizes the current reference range for healthy horses at rest of the related blood parameter. In the case of Albumin, *p*-values were not adjusted to show the evidence of the adjustment. *n* = 55.

**Figure 2 animals-13-03110-f002:**
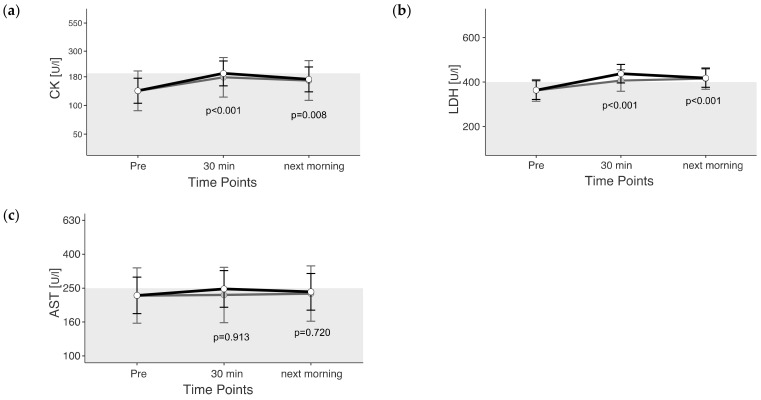
(**a**–**c**) Estimated marginal means (EMM) (white dots) and their 95% confidence intervals (black whiskers) of muscle enzymes at measured time points. Gray dots and whiskers represent EMM and 95% confidence intervals when 10 and 30 min post-exercise values were adjusted for plasma shift. *p*-values apply to the difference between Pre and the respective time point. Y-axis is scaled according to the observed scores. The gray box symbolizes the current reference range for healthy horses at rest of the related blood parameter. *n* = 55.

**Figure 3 animals-13-03110-f003:**
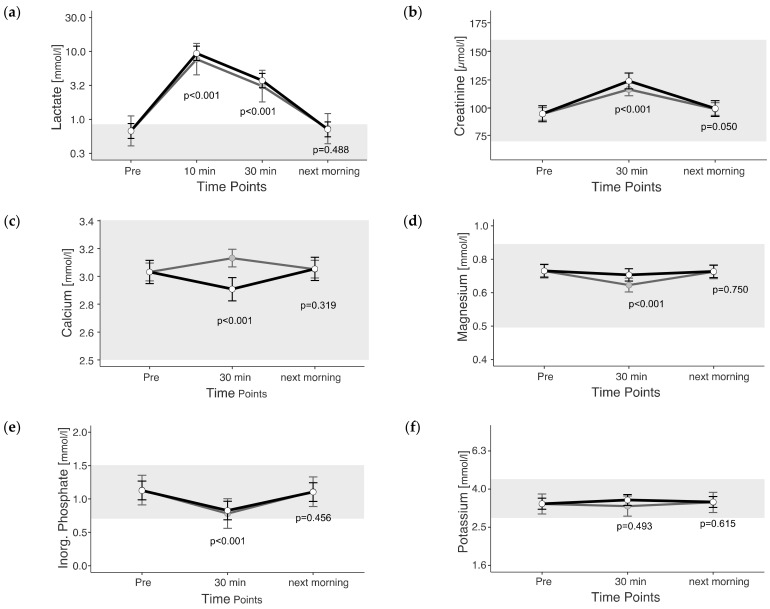
(**a**–**f**) Estimated marginal means (EMM) (white dots) and their 95% confidence intervals (black whiskers) of muscle metabolites at measured time points. Gray dots and whiskers represent EMM and 95% confidence intervals when 10 and 30 min post-exercise values were adjusted for plasma shift. *p*-values apply to the difference between Pre and the respective time point. Y-axis is scaled according to the observed scores. The gray box symbolizes the current reference range for healthy horses at rest of the related blood parameter. *n* = 55.

**Figure 4 animals-13-03110-f004:**
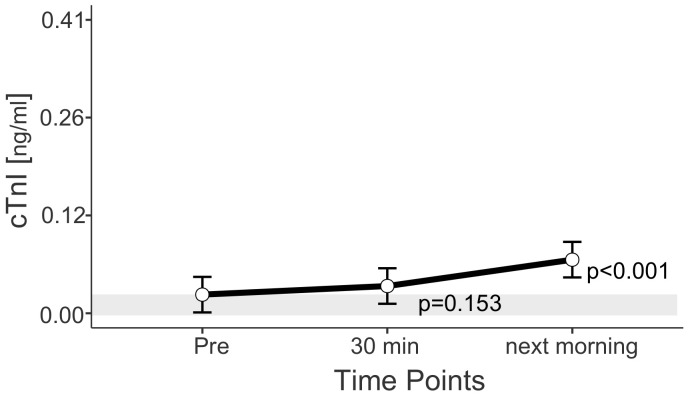
Estimated marginal means (EMM) (white dots) and their 95% confidence intervals (black whiskers) of cTnI at measured time points. *p*-values apply to the difference between Pre and the respective time point. Y-axis is scaled according to the observed scores. The gray box symbolizes the current reference range for healthy horses at rest. Blue dots show the observed values. *n* = 55.

**Table 1 animals-13-03110-t001:** Age, sex and breed of participating horses in cross-country rides.

Horse	Age	Sex	Breed
1	7	Mare	Holsteiner
2	12	Mare	Stud Book du Cheval Selle Français
3	12	Gelding	Hanoverian
4	15	Mare	Hanoverian
5	12	Gelding	Irish Sport Horse
6	14	Gelding	Hanoverian
7	9	Gelding	Hanoverian
8	10	Mare	Polish Horse Breeders Association
9	15	Mare	Rheinlander
10	9	Mare	German Sport Horse
11	7	Mare	Oldenburger
12	8	Mare	Oldenburger
13	8	Gelding	Westphalian
14	14	Gelding	Holsteiner
15	11	Gelding	Hanoverian
16	12	Mare	Hanoverian
17	10	Gelding	Irish Sport Horse
18	11	Gelding	Holsteiner
19	15	Gelding	Hanoverian
20	7	Mare	Hanoverian

**Table 2 animals-13-03110-t002:** Cross-country competition information, as well as distance, speed, environmental parameters, ground and altitude profile.

Competition	Level	Week	Venue	Horses N =	Distance [m]	Speed Required [m/min]	Temperature [°C]	Ground	Altitude Profile	Weather
1	CCI3*-S	12	A	11	3401	550	13	normal	flat	cloudy
2	CCI3*-S	15	B	7	3007	520	16	normal	intermediate	sunny
3	CCI2*-S	16	A	2	3000	520	16	normal	flat	cloudy
4	CIC2*-L	18	C	1	3787	520	16	deep	hilly	sunny
5	CCI2*-S	18	C	2	3085	520	18	normal	hilly	cloudy
6	CCI4*-S	18	C	7	3705	570	20	deep	hilly	sunny
7	CCI4*-S	24	A	3	3772	570	29	normal	flat	sunny
8	CIC3*-L	25	A	3	4455	550	26	normal	flat	sunny
9	CCI3*-S	25	A	3	3364	550	25	normal	flat	sunny
10	CCI2*-S	30	D	2	2661	520	20	normal	hilly	cloudy
11	CCI3*-S	30	D	6	3538	550	20	normal	hilly	cloudy
12	CCI2*-S	35	E	1	3087	520	28	normal	flat	sunny
13	CCI3*-S	35	E	2	3470	550	28	normal	flat	sunny
14	CCI4*-S	35	E	5	3580	570	28	normal	flat	sunny

CCI = Concours Complet International; CIC = Concours International Combiné; “*” = star classification; “-L” = long format; “-S” = short format.

**Table 3 animals-13-03110-t003:** Blood-based biomarker and associated categories.

Category	Biomarker
Fluid Balance	RBC, HCT, TP, Albumin, UREA, SDMA, Chloride, Sodium
Muscle Enzymes	CK, LDH, AST
Muscle Metabolites	Lactate, Creatinine, Inorg. Phosphate, Calcium, Magnesium, Potassium
Cardiac Markers	Cardiac Troponin I (cTnI)

## Data Availability

The data presented in this study are available on request from the corresponding author. The data are not publicly available due to privacy.

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
