# Peer review of "Blood-Based Markers for Skeletal and Cardiac Muscle Function in Eventing Horses before and after Cross-Country Rides and How They Are Influenced by Plasma Volume Shift"

_animals, 2023, doi:10.3390/ani13193110_

Round 1

Reviewer 1 Report

The authors report the results of a large survey of post-exercise blood sample analysis in a relatively under-reported population of equine athletes: eventers performing the cross-country phase of the discipline.  The size of the resulting database, and the fact that the data reflect the recovery from a more “modern” version of the cross-country phase compared to the existing scientific literature, makes the results particularly valuable. A limited number of changes are necessary to provide the reader with a more clear and accurate view of how the data were collected. The majority of the recommended edits center on the interpretation of the data to incorporate current concepts in the field of exercise physiology.

The authors make the broad assumption that fatigue exists in all elite eventing horses after exercise, and that changes in specific blood borne markers illustrate that fatigue.  Unfortunately, neither is sufficiently and consistently true to conclude what the authors would like to conclude (i.e., an increase in serum CK shows that the associated exercise resulted in muscle fatigue). Fatigue in the context of exercise is taken to mean the temporary loss or inability to perform work.  Based on the description of the exercise performed for this study (training gallop), the horses may have been somewhat tired but were unlikely to be fatigued – certainly not in the sense of the common equine exercise physiology use when horses on a treadmill are unable to maintain a defined exercise intensity despite encouragement.  Increased concentrations of analytes normally found inside cells such as CK, LDH, and cardiac troponin suggest a loss of cell membrane integrity for at least some portion of the associated tissue (i.e., damage, as outlined in the introduction), and while that can often be associated with fatigue, it is not predictive.  Therefore, this reviewer strongly recommends that the manuscript be edited to remove most references to “fatigue” (both whole body and tissue-level fatigue, the exception being when referring to published literature in which fatigue was confirmed), and instead describe the exercise challenge used in the study in a more precise context of distance/time/average speed.

Line 17: The term “risk factor” is used to define a pre-existing condition that is associated with a subsequent outcome. As such, the term is incorrectly used by the authors – instead, they likely mean that horses are at risk for fatigue when performing in eventing competitions.

Lines 60-66: More recent definitions of central and peripheral fatigue now include dysfunction of the peripheral nervous system (both sensory and motor neurons) under the category of “peripheral" fatigue, and restrict the term “central” fatigue to the inability of unwillingness of the central nervous system to activate motor neurons. DOMS is not commonly included as a component of fatigue, as DOMS is widely considered a phenomenon of myofiber damage, not exhaustion.

Lines 116-120: There is likely a considerable variation in the relative efforts during the exercise sessions, with some occurring during training and others during competition. Ideally, the authors can report specific work intensities for the various exercise session – at the very least, the data needs to be separated by training vs competition.

Lines 184-186: The mixed model used to analyze the data is somewhat complex and doesn’t necessarily match the collected data as outlined in the Methods. For instance, “horse” and “rider” are distinct variables, but only “horse” has been described in the Methods.  Similarly, the authors should define “competition” and “event” or change these terms to match what has been described in Methods sample collection. The authors should consider presenting the mixed model as an equation to help illustrate how the variables were incorporated.

Lines 238-242: Given that the authors are treating the measured variables as being grouped according to proposed physiological relevance (Table 1), it would likely be more useful to then report them in the same grouping rather than alphabetical order. It is probably not necessary to have BOTH median/quartile and EMM/confidence interval data displayed in a table – in fact, it may not be necessary to have either table given that the figures present the data more effectively.

Figure 1a: Probably shouldn’t display the adjusted values for albumin since those have been specifically manipulated in something of a circular argument (of course it is a straight line – the adjustment formula was specifically designed to produce exactly that).

Line 227: Urea didn’t peak – at least not statistically – until the morning after the ride.

Lines 315-326: See general statement at beginning of the review. It is a very large stretch to conclude that fatigue existed in the horses and/or the individual tissues based on these data.

Section 4.1 Fluid shifts: This section merits some degree of expansion, given that the exercise performed in this study would typically be considered submaximal, at least compared to the maximal exercise challenges used in most of the references cited. At the same time, the exercise intensity was likely greater (and duration shorter) than many of the published studies of fluid shifts in endurance horses.  Thus, the authors present some relatively unique data within the continuum of equine exercise and should more explicitly highlight how the fluid shifts found in this study compare to other studies with different exercise challenges.

Section 4.2 Muscle enzymes: The authors are encouraged to evaluate their selected endpoints in the context of analyte kinetics; i.e., to what extent does the change from one timepoint to the next reflect a change in release into the blood vs a change in clearance. From this standpoint (and considering the superior specificity of CK for skeletal muscle compared to LDH), the evidence would suggest that whatever skeletal muscle insult that occurred during exercise is resolved by the following morning, and the slight persistence of CK in the blood is a function of slow clearance as opposed to continued release (and thus does not reflect the possibility of continued dysfunction in terms of either fatigue or DOMS).

Section 4.3 Muscle metabolites: This section requires a lot of work. The authors need to incorporate a discussion relating to acid-base shifts to help explain some of the acute changes in electrolytes (see recent articles by Lindinger et al – even though blood pH was not specifically measured, most of the components that influence pH were so an educated guess can be offered). This may be particularly helpful in addressing the change in inorganic phosphorus, which in the context of pH, acts as a component of [Atot]. In terms of lactate, the authors should review the lactate shuttle concepts as outlined by Brooks et al to evaluate blood lactate in terms of rate of release by glycolytic fibers vs rate of clearance by liver and oxidative fibers.

A handful of awkwardly-constructed sentences that can likely be addressed by editorial staff.

Reviewer 2 Report

This is an interesting work that aim to determine through some blood markers, muscular and cardiac fatigue in eventing horses. Also, albumin was used to evaluate intravascular volume loss.

Simple summary:

L20: please rephrase. Parameters outside the categories of fluid….

Introduction:

Even though fatigue can occur in these horses, I suggest not including DOMS in the introduction and discussion. Authors mentioned in the introduction that DOMS may occur one to two days after the exercise with mechanical hyperalgesia due to unaccustomed strenuous exercise. Therefore, it does not apply to this study.

Authors could include a bit more information about cardiac troponin evaluation in sport horses to better understand why was included as a fatigue marker.

Material and Methods

More information about horses is needed, like breed, age, weight, general management, etc. Also, information related to training level and level of competition is required.  Clinical and echocardiography was performed by any of the authors? If so, please mention it. Besides being examined before participation, do horses were examined after exercise? for example, was cardiac and respiratory frequencies determined? or time to normal RR, sweat level, etc?

From riders some information would be appreciated, including experience, weight, etc.

Data about ambient conditions during competition (temperature, humidity, etc.) also should be included.

This information is needed to better understand the study design and factors that could affect the results.

Authors could consider rearranging materials and methods section. Start with paragraph L106-113. Then, mention period of time when study was performed (L170-172) explain all the information about horses, riders, competition (level, type, L173-177), conditions, etc. (as mentioned before). For example, L117-120 about competition type should be in this section.

Then, methodology should be detailed: sampling times (L116-117, L120-124), sample collection and processing (L139-167 merging those paragraphs, as appropriate to each step from time sample was obtained until it was processed). Then, blood parameters (L126.137).

Finally, statistical analysis.

Was the adjusted formula validated before? If so, please detail. If not, is there any previous validation study? Otherwise, should be much more explained and justified as a method used to adjust a marker further use to make conclusions.

Results

Paragraph L208-220 should be included in material and methos section.

Sampling time 1 minute after exercise could be left out from results, considering n size is 7 and only a few variables were determined.

It is suggested to rewrite results because it is a bit hard to follow the idea. Maybe describing finding for each variable at a time or by sampling time, but same description for all variables.

As results are presented in graphs, table 2 and 3 could be included as supplementary data, unless authors consider it is fundamental to have that information in the results sections.

Tables should include references regarding reference range for all variables. Also, for sampling time next morning, an hour range could be included, considering those samples were obtained in a wide range of time (L122-124) and may not be 24 hours after first sample. Title of both tables could be a little more self-explicative.

Figure titles should include n size, and explanation of p values (between sampling times, etc.)

Figures 1 c and d does not have the gray box symbolizing reference range.

L253-254: those maximum values correspond to one horse, one sampling time, if so, what sampling time? Because were not included in tables or graphs, therefore it is hard to understand the meaning. Please better detail and explain those values (same for L270, L296).

L254: CK reached values above the reference in 54% of the rides, LDH in 65% and AST in 41% of the rides. At what sampling time? Also, were those rides in all competition types (based on the star level)? or were observed in the more difficult ones? Please explain.

L256-266: please change that CK and LDH values were expected to increase above reference values. This is more an ¨expected¨ result than a result by itself (redaction).

Figure 2 title should only include pre. 30 minutes and next morning (1 and 10 minutes were not included, L265-266).

L272-273: are those reference values different to those included in table 2 and 3? because are different. Please modify and use the same reference values in the manuscript. Also, in L270-272 it is stated that electrolytes and creatinine were close to reference values, except for potassium, but based on figure 3, all variables were within each reference value. Please explain.

In addition, it would be interesting to include in the discussion why authors mention in result section when a specific variable was increased or decreased in X% of rides (e.g., L274-275 for potassium).

L276: potassium showed no significant (instead of relevant).

Do authors examine possible differences regarding the type of competition? For example, 9 rides in a 2* competition, 31 rides in a 3* competition, 15 rides in a 4* competition, 1 ride in a VA, and 7 rides in training. Was there in difference in blood variables considering the competition level? If so, please include.

Discussion

In general, it is suggested to enrich discussion with more horse-related publications. Also, when human studies are included, please mention that are in humans, otherwise the reader tend to imply that are in horses.

Specific comments for the discussion:

The first paragraph should be rewritten because contradicts the first part with the second. Also, please avoid using words/phrases like relevant, extent of peripheral fatigue, drops to very low level, reflecting the extent, etc.

L334: that study refers to 47 or 48?

L377-382: should be included in the result section.

L397-403: the mentioned study (59) is in Arabian racehorses, not Thoroughbred.

L397-398: are those results comparable? if so, please better explain.

L399: why authors think LDH values didn´t drop the next morning?

L400-402: it is not possible to establish that results from the present study were significantly lower than those of Kowalik and Tomaszewska study (same for L498-499). Please modify.

L404-408: please include more discussion (and include reference/s used if applicable) about why authors think eventing horses were able to better metabolize lactate with equal LDH increase, implying a better training status. Training level is important to include in material and methods section because of this…it must be included as part of the discussion.

L409-417: please consider including more horse studies in this paragraph.

L418-422: mention references used.

L426-428: why authors think trained eventing horses have higher AST levels compared to average horses (what is the definition for an average horse?...training, use, sport, etc.?).

L429-434: same as L426-428, please better discuss why authors are implying in this paragraph. What is s.o.?

L449: inorganic phosphates drop after 30 minutes (sampling time presented in results) not immediately after exercise. Please modify.

Again, why was relevant to mention %rides where some variables were increased or decreased? Please discuss this.

L453-457: please explain with more detail this paragraph or hypothesis.

L462-473: DOMS section should be excluded from the discussion (and introduction) considering it is not part of the objectives, and it not related to the study.

L476-477: why authors stated that increased cTnI in eventing horses indicates cardiac fatigue? please include more references that allow to conclude that. Some previous studies indicate that much higher cTn values should be determined to think about a possible cardiac fatigue or cardiac problem.

L491: maximum values were obtained when, at what sampling time, in one horse? please explain.

As a suggestion, cardiac markers section could also include other possible factors for different values obtained in the present study compared to previous. For example,

method to detect the marker, type of competition, type of horse, emphasize when other studies report higher/lower values when using horses with cardiac disease, etc. Authors could review studies like Shields et al 2018 or Lippi & Sanchis-Gomar 2020, for example.

It is appreciated that authors include limitations, but some should be included in material and methods section because are relevant for the study design.

Manuscript should be revised because grammar errors were detected. 

Also, please consider using more scientific terminology in some specific parts of the manuscript (mentioned in comments).

Reviewer 3 Report

The study's concept is commendable and necessary, considering the declining social acceptance of sport horse due to insufficient research delivering methods to prevent injuries and enhance welfare.
This study aimed to investigate whether blood-based markers for peripheral and cardiac fatigue are useful in eventing horses before and after cross-country rides and how they are influenced by plasma volume shift. Thus, the approach of the study appears original. The contents of the manuscript are quite interesting because of his methodology and the tools of quantification used. I find it interesting. I thus find that this paper definitively delivers results that will surely be of interest to the readership of the journal Animals.

However, there are concerns regarding the superficial description of the methodologies, which needs some corrections. Also in some parts the article seems to be chaotic.

Introduction

“As fatigue cannot be assessed by a single indicator alone, compiling a panel of exercise-dependent biomarkers and establishing individual reference ranges has helped to achieve this assessment of recovery status in high-performance human sports [3-11].” I am really glad about that statement however in my opinion it should be expanded. The Authors should add some information about the physical activity-dependent hematological and biochemical changes and several novel biomarkers such as anti-inflammatory cytokines which have been proposed as markers for exercise monitoring in humans and horses (ex. IL-ra, IL-13). In my opinion, there is no presentation about blood-based biomarkers which were evaluated in previous studies.

Materials and methods

In my opinion, the explanation of which parameters delayed transport to the laboratory to perform measurements should be added. 48 hours from blood collection sometimes is to a long time to obtain various messurments.

Samples collection and blood collection paragraph should be together because it is really hard to understand the procedure if it is written this way.

Results

The information about horses age should be added to the Materials and Methods section.

Discussion

The physical activity-dependent hematological and biochemical changes in horses of different types of usage should be disused (ex. race, endurance, leisure), because it strongly influences on basal as well as after-exercise values of these parameters. There is no discussion connected with white blood cell reaction. Woth mentioned is that intense exercise influences on horses' hormonal reaction ex. cortisol concentration, as well as testosterone to cortisol ratio in sport horses as well as in leisure horses. Hormones also influence WBC (ex. cell margination) as well as RBC and HT.

The main limitation of this study was the number of horses, which was too small to establish reference values according to the American Society for Veterinary Clinical Pathology’s guidelines for the determination of reference intervals in veterinary species. Thus it should be added to the limitation section and discussed.

Minor corrections should be added

Round 2

Reviewer 2 Report

Manuscript has improved and suggestions were incorporated which is appreciated.

P2L80 and 83: Arabians and Thoroughbred

Table 1. Age, sex, and breed of participating horses in cross-country rides.

Table 2. Cross-country competition information, as well as distance, speed, environmental parameters, ground, and altitude profile.

P5L162-163: time span between 30 minutes sample and next-day was between 11 to 21 hours (10 hours difference). Do authors consider that some results could have been different without this 10-hour difference?

P5L172-173: Lactate values were determined…move to 2.2.1 section.

P5L190: (Table 3).

P5L196: Table 3.

P6L227: plasma volume loss, and dehydration were included (?).

Results: authors could consider mentioning when are describing data without and with adjustment, it is somehow confusing to understand at some points.

P8L298: Figures 1 a-h: Estimated marginal means (EMM) (white dots) and their 95% confidence intervals (black whiskers) of fluid balance blood parameters at measured time points. Gray dots and whiskers represent EMM and 95% CI when 10 and 30 minutes post-exercise values were adjusted for plasma shift…..

P9L308: Figures 2 a-c: Estimated marginal means (EMM) (white dots) and their 95% confidence intervals (black whiskers) of muscle enzymes at measured time points. Gray dots and whiskers represent EMM and 95% CI when 10 and 30 minutes post-exercise values were adjusted for plasma shift…

P10L342: Figures 3 a-f: Estimated marginal means (EMM) (white dots) and their 95% confidence intervals (black whiskers) of muscle metabolites at measured time points. Gray dots and whiskers represent EMM and 95% CI when 10 and 30 minutes post-exercise values were adjusted for plasma shift…

P11L368: Blue dots show the observed values (no blue dots are shown)..maybe white dots?

No adjustment was made with cTnI, and no information about that is in materials and method section.

P11L382-386: Even though cross-country ride represents a submaximal exercise, a 17.9% HCT increase 10 minutes after exercise was observed in horses from the present study, suggesting that either spleen storage capacity is even greater (then what? or compared to?), or that HCT is additionally increased by plasma volume loss. (or similar redaction).

P11L392-394: it would be interesting to mention the type of exercise included in Masri et al and McKeever et al studies for q better comparison with the present study results.

P12L413-416: please rephrase (these parameters refers only to fluid balance metabolites or all?; EMMs of these parameters 10 and 30 min after exercise were unchanged compared to pre and the changes were not statistically significant…please rephrase because it´s not easy to understand what authors are concluding).

P13L454: Un the present study, LDH values 30 min after exercise (LDH…) were about the same as those reported in Purebred Arabian racehorses (LDH…) in daily training (2000 m gallop) (49). L460-461: maybe merge with this paragraph.

L457-459: Lactate values….maybe this phrase could be moved to muscle metabolite section (4.3).

P13L469-471: authors could consider mentioning why CK values decreased faster than human athletes to better contextualize using that study in humans.

P13L492-497: consider moving this paragraph up where CK and LDH are discussed.

P13L486-491: these phrases could be merged and summarized with the previous information, mainly because they discuss why AST is increased in trained horses, but not particularly results from this study, where AST did not increase significantly.

P14L503: please consider change extreme.

P14L506-507: maybe include more information about creatinine and the increase 30 min after exercise.

P15L565: in racehorses 2 h

P15L568: which is more than 20-fold higher than the maximum value…

P15L571-573: consider merging this paragraph with the previous or better include this information, because it mentions mean increases calculated in 2 studies, but is not related to the present study. Same with L574-579, it´s confusing to read those phrases are relate them with the results/discussion, please rephrase.

P16L623-624: urea, CK, LDH and cTnI were significantly, while the other parameters returned to pre-exercise values.

Do authors consider that albumin adjustment achieved the objective? if so, a comment could be included in conclusions.

Please change are to were in: P7L277, P7L284, P7L290, P9L319, L390, L323, L327, L330, P10L339, P11L374, P12L413, L415-416, L431, P13L455, L473, L488, P14L508, L539, P15L562, P16L624.

Some changes and correction must be included.

Reviewer 3 Report

After corrections the quality of the article increased significantly. Now It can be published in the Journal
